# Neutrophil Interactions with the Lymphatic System

**DOI:** 10.3390/cells10082106

**Published:** 2021-08-17

**Authors:** Arnolda Jakovija, Tatyana Chtanova

**Affiliations:** 1Innate and Tumor Immunology Laboratory, Immunity Theme, Garvan Institute of Medical Research, Darlinghurst, NSW 2010, Australia; a.jakovija@garvan.org.au; 2St Vincent’s School of Medicine, Faculty of Medicine, UNSW Sydney, Sydney, NSW 2052, Australia; 3School of Biotechnology and Biomolecular Sciences, Faculty of Science, UNSW Sydney, Sydney, NSW 2052, Australia

**Keywords:** lymphatic system, neutrophils, inflammation

## Abstract

The lymphatic system is a complex network of lymphatic vessels and lymph nodes designed to balance fluid homeostasis and facilitate host immune defence. Neutrophils are rapidly recruited to sites of inflammation to provide the first line of protection against microbial infections. The traditional view of neutrophils as short-lived cells, whose role is restricted to providing sterilizing immunity at sites of infection, is rapidly evolving to include additional functions at the interface between the innate and adaptive immune systems. Neutrophils travel via the lymphatics from the site of inflammation to transport antigens to lymph nodes. They can also enter lymph nodes from the blood by crossing high endothelial venules. Neutrophil functions in draining lymph nodes include pathogen control and modulation of adaptive immunity. Another facet of neutrophil interactions with the lymphatic system is their ability to promote lymphangiogenesis in draining lymph nodes and inflamed tissues. In this review, we discuss the significance of neutrophil migration to secondary lymphoid organs and within the lymphatic vasculature and highlight emerging evidence of the neutrophils’ role in lymphangiogenesis.

## 1. Introduction

Lymphatic vessels serve as a drainage system for collecting and returning interstitial tissue fluid and protein (lymph) to the bloodstream [1]. In addition, lymphatics provide an important pathway for leukocytes during an immune response [2]. On the flip side, immune cells can regulate lymphatic vessel development, growth and function. Recruitment and activation of immune cells in the context of inflammation promotes lymphangiogenesis and also increases lymph flow [3]. These effects amplify cell trafficking to lymph nodes and are important for mounting an effective immune response.

Under homeostatic conditions, neutrophils patrol the lymph nodes that drain the skin, lungs and gastrointestinal tract [4]. They enter the lymph nodes via high endothelial venules (HEVs) in an L-selectin-dependent manner and reside within the lymph node interstitium [5]. Neutrophil lymph node egress via efferent lymphatics is mediated by sphingosine-1-phosphate receptor (S1PR) [5].

At the onset of inflammation (e.g., infection, tissue damage, and cancer), neutrophils rapidly migrate to sites of injury, where they can destroy microorganisms through phagocytosis, release anti-microbial molecules and promote immune cell recruitment [6]. Once neutrophils enter inflamed tissues, they have several possible fates: most neutrophils are thought to die at the site of inflammation; however, a proportion can reverse transmigrate back into blood vessels [7,8]. Furthermore, some neutrophils can egress sites of inflammation by entering lymphatic vessels and migrating to draining lymph nodes. This lymphatic migration of neutrophils has been observed in response to bacterial and parasitic infections and after vaccination [9,10,11,12].

In addition to the lymphatic route, neutrophils can also enter draining lymph nodes from circulation via HEVs (Figure 1). Neutrophil entry into the lymph node through HEVs has been observed following intradermal injection of *S. aureus* [13] and Complete Freund’s Adjuvant (CFA) [11] as well as in a model of tumour cell lysis in murine skin [14]. A recent study showed that the majority of neutrophils were recruited to the lymph node directly from HEVs [15], while a smaller population of neutrophils migrated directly from the site of inflammation to lymph nodes via afferent lymphatics [15].

Lymph node neutrophils are phenotypically distinct from circulating neutrophils and have the capacity to modulate adaptive immunity [4]. Although the existence of bona fide neutrophil subsets remains to be conclusively demonstrated, the two different routes of neutrophil entry (via afferent lymphatics or HEVs) suggest that the two populations of lymph node neutrophils may have distinct functions. Neutrophils entering the lymph node via afferent lymphatics are arriving from sites of inflammation, are highly activated, express costimulatory molecules and carry tissue antigens [9,10]. This may enable them to act as antigen presenting cells and to shuttle pathogens to draining lymph nodes. On the other hand, neutrophils arriving from circulation are likely coming from the bone marrow (although it is possible that a proportion of tissue neutrophils undergoes reverse transmigration to enter circulation [7,8] and then migrates to draining lymph nodes) in response to pathogens in the lymph nodes. It is likely that neutrophils entering from circulation play a prominent role in preventing the systemic dissemination of bacteria [15].

Another role of neutrophils in interacting with the lymphatic system is to contribute to lymphangiogenesis, a process which can modulate inflammation by controlling cell trafficking and antigen transport [3,16]. Neutrophils mediate this function by releasing lymphangiogenic molecules, such as VEGF-D, at the site of inflammation [17,18]. Here, we review neutrophil trafficking within the lymphatic system and the effect on the immune response, highlighting new evidence of neutrophils’ role in lymphangiogenesis.

## 2. The Lymphatic Vasculature

The lymph is a biological fluid, whose function is to return the interstitial fluid from tissues to the blood circulation [19]. Since lymph is produced as the result of a capillary filtration process, its composition is very similar to plasma ultrafiltrate, with high numbers of leukocytes and plasma proteins [20]. In addition, lymph contains a large amount of dietary fats, related lipoproteins and cholesterol absorbed from the intestine [21].

Lymph flows through the lymphatic vessels, which form a unidirectional network of vessels that collect the lymph and then return it through a system of collecting vessels, lymph nodes and lymphatic trunks into the venous circulation [22,23]. The initial lymphatic vessels are composed of a single layer of overlapping oakleaf shaped lymphatic endothelial cells, which express the lymphatic endothelial marker podoplanin and the lymphatic vessel endothelial hyaluronan receptor 1 (LYVE-1) [24]. The primary lymphatic valve system is composed of endothelial cells that interdigitate and form loose discontinuous buttonlike junctions (buttons) [25]. The afferent lymphatics start as blind-ended capillaries that branch and merge with larger lymphatic collecting vessels (which downregulate the expression of LYVE-1). Collecting lymphatic vessels are uniquely adapted to ensure leakage-free transport of lymph [26]. They contain valves which ensure the unidirectional flow of lymph and prevent backflow [27]. Collecting LECs have an elongated morphology, are connected by continuous and tight zipperlike cell–cell junctions [25] and enclosed by a continuous basement membrane covered by smooth muscle cells [28]. LECs are exposed to peripheral extracellular antigens after tissue damage, infection or inflammation [29] and can take up these antigens and present them to T cells [30,31]. Afferent collecting vessels merge with the collagen-rich capsule of the draining lymph node and empty their contents into the subcapsular sinus, before exiting as efferent vessels that reconnect to the venous blood [2] (Figure 2).

The lymphatic network has three main functions: (1) balance fluid homeostasis; (2) absorb dietary fat in the intestine and transport it back into the blood stream; and (3) facilitate host immune defence [34]. Furthermore, lymphatic vessels are an essential compartment for immune cells, providing a route for antigen transport and the trafficking of dendritic cells (DCs), T cells and neutrophils to draining lymph nodes during immune activation [35]. Moreover, lymphatic vessels are important for macrophage clearance during the resolution phase of tissue inflammation and infections [2] and for the maintenance of peripheral immune tolerance [3]. Notably, the lymphatic network can also be used by microbial pathogens for dissemination and host colonization [36]. Thus, the lymphatic network modulates the immune response directly, by controlling antigen transport by immune cells to draining lymph nodes, or indirectly, by shaping the lymph node microenvironment.

## 3. Neutrophil Migration to Lymphoid Organs

Neutrophil interactions with the lymphatic system include entry into and migration within lymphatic vessels, recruitment to lymph nodes (via both lymphatic and hematogenous routes) and modulation of lymphangiogeneis in tissues and lymph nodes. This section discusses how neutrophils are recruited to lymph nodes.

### 3.1. Neutrophil Migration via Lymphatic Vessels

Early cannulation studies of afferent lymphatic vessels conducted in sheep and humans under homeostatic conditions detected low number of neutrophils in the afferent lymph [37,38,39,40,41]. Neutrophil numbers increased dramatically but transiently in ovine afferent lymph following vaccination with liposomes containing an innate cell stimulator poly(I:C) and diphtheria toxoid (DT) [42]. These studies provided some of the earliest evidence for neutrophil migration in lymphatic vessels.

Recently, in vivo labelling [9,10] and advances in intravital imaging technologies [12,43] allowed the tracking of neutrophils migrating from the periphery to draining lymph nodes and the visualization of their interactions with the lymphatic vessels in mice [9,12]. In a mouse model of cremaster muscle inflammation, antigen sensitisation led to a rapid transient entry of tissue neutrophils into lymphatic vessels and subsequent crawling along the luminal side of the lymphatic endothelium [12]. Our group observed that neutrophils crawling in lymphatic vessels carried fluorescently labelled bacteria [9]. Moreover, using in situ photolabeling, we showed that neutrophils were rapidly recruited to inflamed skin in response to both microbial and sterile challenges, but emigrated to draining lymph nodes via lymphatic vessels only in the presence of an infectious but not a sterile stimulus [9]. This suggests that neutrophil egress from inflamed tissues may depend on neutrophil phenotype and tissue environment. Additional conditions that promote neutrophil entry into lymphatics vessels are yet to be defined.

Neutrophils enter the lymphatic endothelium through a mechanism named transmigration [44,45] (Figure 3). Rather than migrating through established cellular junctions in the lymphatic endothelium, neutrophils induce the enzymatic degradation and the retraction of LECs [45]. In this way they create large openings that allow their rapid passage through the endothelium and provide a route for further neutrophil migration. This mechanism of neutrophil migration may explain the rapid kinetics of neutrophil recruitment in the lymph nodes, which was found to peak after 4–6 h following the administration of an inflammatory stimulus [9,46].

Neutrophil entry from inflamed tissues into afferent lymphatics involves β2 integrins [44]. Indeed, adhesion molecules, such as ICAM-1 and VCAM-1 (expressed on LECs), can support human and mouse neutrophil lymphatic interactions via binding to leukocyte β2 integrin [44]. Integrin-mediated neutrophil adhesion to inflamed LECs through ICAM-1 binding induces the release of matrix metalloproteinases MMP8 and MMP9 and secretion of 12-hydroxyeicosatetraenoate (12(S) HETE) [2]. Neutrophil migration in lymphatic vessels also requires CD11b (or integrin alpha M, which binds to integrin β2 to form complement receptor 3 [47]), since the inhibition or complete removal of CD11b inhibited neutrophil migration from the skin to draining lymph nodes [9].

Neutrophil lymphatic migration is guided by chemotactic stimuli. Chemokine receptor CCR7 regulates entry of DCs, monocytes and T cells into lymph nodes [51,52,53]. CCR7 ligands, CCL19 and CCL21, are produced by lymph node stromal cells and DCs (CCL19) and by lymphatic endothelial cells (CCL21) [54]. Neutrophil migration to draining lymph nodes also appears to rely on CCR7 in some settings [46]. For instance, in models of TNF-α- and/or Complete Freund’s adjuvant (CFA)-induced inflammation in the skin [46] or cremaster muscle [12] neutrophil migration to lymph nodes was CCR7-dependent. Neutrophils upregulated CCR7 upon entry into inflamed tissues, suggesting that neutrophil priming is required for enhanced chemokine receptor expression [12]. On the other hand, neutrophil egress from the skin and entry into the lymph nodes was independent of CCR7 after *S. aureus* administration to the skin [9]. These opposing outcomes suggest that the requirement for CCR7 is stimulus-dependent and may be influenced by neutrophil phenotype, activation status, tissue microenvironment and other factors. This also indicates that additional chemokine pathways can mediate neutrophil lymphatic migration [9]. Consistent with this hypothesis, inhibition of chemokine receptors CXCR2 and CXCR4 demonstrated that CXCR4, but not CXCR2, was required for neutrophil migration from the inflamed skin to draining lymph nodes [9].

### 3.2. Neutrophil Recruitment to Lymph Nodes via HEVs

In addition to entering lymph nodes via afferent lymphatics, neutrophils can also be recruited there from blood via HEVs. Neutrophil migration into lymph nodes across HEVs was reported for the first time in 1959 [55]. Imaging of inguinal lymph nodes by intravital microscopy following *S. aureus* infection confirmed that local immunization or infection rapidly recruits neutrophils from HEVs to lymph nodes [13]. Many of the neutrophils recruited to the immunized lymph node were mobilized from the bone marrow [13]. Neutrophils can enter lymph nodes via HEVs during infection, antigen sensitization and following immune complex activation [56].

While the molecules required for lymphocyte migration through HEVs are well defined [57], the mechanisms of neutrophil entry via HEVs still need to be elucidated. A recent study of *S. aureus* infection, reported that, similarly to lymphocytes [58], L-selectin on neutrophils can bind glycoproteins, such as peripheral node addressins (PNAds), to allow tethering and rolling, which promote rapid adhesion and emigration (Figure 3) [15]. L-selectin recognizes a family of mucinlike glycoproteins (such as CD34, glycosylation-dependent cell adhesion molecule 1, GLYCAM1, podocalyxin, endomucin and nepmucin), which are heavily sulphated, fucosylated and sialylated when expressed by the HEV endothelial cells of peripheral lymph nodes [59]. Blocking L-selectin reduced neutrophil recruitment to lymph nodes following injection with *S. aureus* [9], CFA [11] or tumour cell lysis [14]. Neutrophils can also use P-selectin glycoprotein ligand-1 (PSGL-1) as well as integrins macrophage-1 Ag (Mac-1) and lymphocyte function-associated Ag-1 (LFA-1) to cross HEVs [11]. LFA-1 mediates neutrophil adhesion and Mac-1 facilitates subsequent crawling in inflamed blood vessels [50]. The CXCR4/CXCL12 axis is also important for neutrophil entry into the lymph nodes via this route [11]. Neutrophils can upregulate CXCR4 after antigen challenge and this allows them to enter the lymph node [11]. Notably, neutrophil recruitment to lymph nodes through HEVs can occur independently of CCR7 [9,11,15]. However, it was found to be highly dependent on complement, especially C5a [15]. Taken together, these findings suggest that neutrophil migration to lymph nodes via afferent lymphatics and through HEVs is regulated by partially overlapping but distinct mechanisms. However, molecular pathways regulating neutrophil entry into lymph nodes via HEVs require further investigation.

## 4. Neutrophil Functions within the Lymphatic System

### 4.1. Pathogen Control versus Dissemination

Neutrophils possess multiple mechanisms for killing pathogens including: (1) phagocytosis; (2) production of highly toxic reactive oxygen species (ROS) in the pathogen-containing vacuole; (3) degranulation; and (4) generation of neutrophil extracellular traps (NETs) [60]. It is likely that these neutrophil functions play a significant role in pathogen control in the lymph nodes. This is important since the lymph nodes are one of the first sites for pathogen spread following infection. The role of neutrophils in limiting pathogen spread has been demonstrated in a model of *S. aureus* infection in mice, where neutrophils were directly recruited to the lymph nodes via the HEVs and provided critical protection, ensuring that no bacteria reached the circulation [15]. A combination of intravital imaging and in vitro microfluidic device models revealed how large numbers of neutrophils navigate through capillary networks and interstitial spaces in organs and tissues, such as lymph nodes [61]. The perturbation of chemoattractant gradients and increased hydraulic resistance induced by the first neutrophil in one capillary branch alters the migration of the following neutrophil towards the other branch, and this helps distribute neutrophil traffic uniformly at bifurcations of capillaries [61]. These mechanisms may promote efficient neutrophil trafficking during infections and inflammation and help avoid bacterial dissemination [61].

Neutrophil localization in lymph nodes may reflect their role in pathogen containment and immune regulation. For instance, adoptive transfer of neutrophils into afferent lymphatic vessels showed that lymph derived neutrophils localized in the subcapsular or medullary sinus of the lymph node [62]. This positions neutrophils close to the sites of pathogen entry. Here, neutrophils may interact with subcapsular sinus macrophages to prevent pathogen spread [63]. Moreover, a study of LECs in human lymph nodes by single-cell RNA sequencing showed that LECs lining the floor of the subcapsular and medullary sinuses constitutively expressed neutrophil chemoattractants, such as C-type lectin CD209 [64]. CD209+ LECs may cooperate with medullary sinus macrophages to recruit neutrophils into this region during infections [64]. Notably, a small population of neutrophils may be positioned in the parenchyma of lymph nodes under basal conditions awaiting disseminating bacteria. These findings highlight a crucial role for neutrophils in preventing bacterial spread from draining lymph nodes [15].

On the other hand, if a pathogen is able to survive within neutrophils, then the neutrophils can become “trojan horses” concealing pathogens from other immune cells and transporting them from the periphery to secondary lymphoid organs [65,66]. In an intradermal infection model of *Mycobacterium bovis* strain Bacille Calmette–Guérin (BCG), neutrophils migrated through lymphatic vessels and accumulated in large numbers in the subcapsular sinuses of lymph nodes while still harbouring *Mycobacteria* [10]. Likewise, GFP-tagged *Salmonella abortusovis* was also shuttled by neutrophils from the oral mucosa of infected sheep to lymph nodes [67]. There, neutrophils could shelter pathogens from other immune cells and impair or delay the induction of the adaptive immune response. Moreover, following infection with obligate intracellular protozoan *Leishmania* [68], neutrophils were the main subset containing viable parasites while neutrophil depletion decreased parasite burden. Likewise, *S. aureus* survival inside neutrophils contributed to infection [69]. In *Toxoplasma gondii* infection, neutrophils may maximize the probability of parasite transmission by promoting an effective immune response that facilitates both host survival and establishment of persistent infection [70]. Together these studies highlight a dual role for neutrophils in secondary lymphoid organs. On the one hand, neutrophils are important for pathogen containment in draining lymph nodes. On the other hand, some pathogens are able to exploit this response against infection and have evolved mechanisms to benefit from the early neutrophil influx and evade detection by other immune cells.

### 4.2. Antigen Presentation

Neutrophils can ferry antigens from the site of infection to the draining lymph node. At the early timepoints they are the first immune subset migrating to draining lymph nodes [9,10,71,72]. This suggests that neutrophils are important contributors to the regulation of early adaptive immune responses against foreign antigens or infectious agents, following their rapid migration in the lymphatic system [10,71].

They can exert this function directly or indirectly: directly, by secreting immunomodulatory molecules that recruit and activate leukocytes and by interacting with lymphocytes and presenting antigens; indirectly, by facilitating the transport of pathogens and antigens to lymph nodes in order to activate resident DCs and macrophages, which can then present antigens to lymphocytes [72,73]. Antigens are transferred to lymph node resident DCs and macrophages through phagocytosis of neutrophil derived vesicles (apobodies, exosomes or microvesicles) [56]. Neutrophil depletion in a mouse model of *M. tuberculosis* infection increased the frequency of *M. tuberculosis*-infected DCs in the lungs, but decreased the trafficking of DCs to the lymph node. This led to delayed activation and proliferation of *M. tuberculosis*-specific CD4 T cells, suggesting that neutrophils can enhance DC maturation and antigen presentation [74]. Notably, neutrophil-DC hybrids with neutrophil and DC properties have been detected during inflammation [75]. They have been observed in pre-clinical murine models of fungal infection (blastomycosis, aspergillosis and candidiasis) [75]. These cells not only expressed both neutrophil and DC markers, but also presented foreign protein antigens to naïve CD4 T cells [76]. These hybrid cells have been detected in the peritoneal cavity, skin, lung, and lymph nodes under inflammatory conditions [77].

### 4.3. Neutrophil Modulation of T and B Lymphocytes

Neutrophils and T cells colocalise in lymph nodes and tissues in steady state and in inflammation [9,78]. Several reports have shown that neutrophils can present antigens to T cells, activate them and enhance T cell proliferation [79]. On the other hand, neutrophils can also suppress T cell proliferation and induce apoptosis [80]. Thus, it is important to understand the mechanisms behind neutrophil–T cell interactions. T cells require two signals for activation: antigen recognition via the T cell receptor (TCR), and a costimulatory signal provided by interactions between CD28 on T cells and B7 family members (CD80, CD86) expressed on antigen presenting cells (APCs) [81]. Human neutrophils can function as APCs and activate CD4+ T cells in vitro and ex vivo [79]. Similarly, neutrophils can also activate CD8+ T cells via MHC class I antigen presentation [82]. Indeed, neutrophils can express molecules associated with antigen presentation, including MHC class II and costimulatory molecules [9], after exposure to granulocyte-macrophage colony-stimulating factor (GM-CSF) or interferon-γ [83].

In different settings neutrophils can inhibit T cell function [84]. Neutrophils execute this function by releasing ROS, NO or Arginase 1 (Arg-1) in close proximity to target lymphocytes [85]. This can lead to downregulation of TCRζ on T cells, thereby arresting cells in the G0-G1 phase [86]. Neutrophils can also suppress T cell proliferation through the release of hydrogen peroxide in the immunological synapse [86]. After stimulation with IFN-γ, neutrophils can express the immune checkpoint inhibitory molecule programmed death ligand (PDL)-1 [87]. Upregulation of this ligand is associated with interferon-dependent PD1-mediated T-cell apoptosis [88]. The outcome of neutrophil-T cell interactions may depend on the tissue microenvironment and neutrophil phenotype (including whether neutrophils are tissue-experienced and antigen-loaded or recruited directly from the bone marrow).

Neutrophils can also directly modulate the B cell response by secreting cytokines, such as B-cell-activating Factor of the tumour necrosis factor family (BAFF) and a proliferation-inducing ligand (APRIL) [89]. BAFF (also known as TNFSF13B) is important for B cell survival and maturation [90]. APRIL (or TNFSF13), secreted by recruited neutrophils in mucosa associated-lymphoid tissue (MALT), can promote the survival of plasma cells [91]. Neutrophils are not usually detected in the germinal centres of lymph nodes; however, a subpopulation of neutrophils was found in the perifollicular areas and marginal zone of the spleen [92]. Splenic sinusoidal endothelial cells are located near perifollicular neutrophils and secrete IL-10, which promotes neutrophil B cell helper function [93]. These B-helper neutrophils induced T-independent production of IgG and IgA by secreting BAFF, APRIL, CD40L and interleukin-21. Moreover, neutrophils are an important source of NETs in lupus, which can directly activate memory B cells through TLR9 stimulation, leading to anti-NET antibody production [94]. Taken together, these studies provide compelling evidence of the importance of neutrophil lymphatic migration to the regulation of the adaptive immune response.

### 4.4. Neutrophil Contributions to Lymphangiogenesis

Emerging evidence suggests that neutrophils may also interact with the lymphatic system by regulating lymphangiogenesis (or growth of new lymphatic vessels) in inflamed tissues and lymph nodes during inflammation and cancer progression [3,95]. Recent studies have identified signals which promote the formation of the lymphatic vasculature [96,97]. Vascular endothelial growth factors, such as VEGF-A, VEGF-C and VEGF-D can induce LEC sprouting and proliferation. VEGFs bind with different specificity to three endothelial transmembrane receptors: VEGFR-1, VEGFR-2, and VEGFR-3. The best characterized axis in lymphatic expansion and development involves VEGFR-3 and its ligands, VEGF-C and VEGF-D [98].

The functional consequences of inflammation-induced lymphangiogenesis require further investigation. In some settings, expansion of the lymphatic network may support inflammation by promoting the transport of immune cells and antigens to the lymph node and enhancing the immune response. In other settings, lymphatic vessel expansion contributes to resolution of inflammation by draining inflammatory mediators and cells from the site of inflammation [99]. For instance, inhibiting the lymphatic vasculature led to increased inflammation in mouse models of skin inflammation [100,101], inflammatory bowel disease [102,103] and rheumatoid arthritis [104]. However, inhibiting this process improves graft survival in corneal transplantation [105,106].

Unlike the well-established role for neutrophils in promoting blood vessel formation (angiogenesis) by producing VEGFs [107], neutrophil contribution to lymphangiogenesis is only just beginning to be uncovered [22]. Neutrophils secrete VEGF-D at sites of inflammation, which promotes proliferation of LECs, stimulates lymphangiogenesis [108] and plays an important role during the development of the lymphatic system [109]. Neutrophils may also increase VEGF-A bioavailability [18]. In fact, after secretion VEGF becomes bound to the ECM and the interaction between VEGF-A and matrix proteins is mediated through the carboxy terminal region, also known as the heparin-binding or ECM-binding domain [110]. This domain is able to bind heparin sulphate proteoglycans and other matrix proteins [110,111] and is thought to be responsible for the sequestration of VEGF-A within the matrix. Neutrophils act via MMP-9 and heparanase (which cleaves heparan sulphate preoteoglycans (HSP) side chains and releases the HSP-trapped VEGF-A) (Figure 4) to increase the amount of biologically active VEGF-A, which in turn activates inflammatory lymphangiogenesis [18].

Neutrophils may also promote tissue lymphangiogenesis by cooperating with macrophages [18,112], as both macrophages and neutrophils contributed to inflammatory lymphangiogenesis in zebrafish by expressing VEGF-A, VEGF-C and VEGF-D [113]. Neutrophils also expressed VEGF-D in the mucosa and lumen of inflamed airways [17]. Furthermore, a study of lymphangiogenesis in abdominal aortic aneurysm (AAA) reported neutrophil infiltration around lymphatic microvessels [114].

Similarly to tissue lymphangiogenesis, VEGF-A-VEGFR2 and VEGF-C/D-VEGFR3 are the principal mediators of lymph node (or intranodal) lymphangiogenesis [115]. During inflammation, B cells [116], macrophages recruited from peripheral tissues [117] and fibroblast-type reticular stromal cells [118] are the major sources of VEGF-A in the lymph node. Emerging evidence suggests that neutrophils also contribute to lymph node lymphangiogenesis [18]. In mice lacking B cells, neutrophils compensate to support lymph lymphangiogenesis during prolonged inflammation [18]. However, more studies are needed to determine whether specific neutrophil subsets are involved in this process.

Together, these studies support the emerging concept of neutrophils as important contributors to lymphangiogenesis in inflamed tissues and draining lymph nodes via the secretion of lymphangiogenic factors and LEC stimulation during the early and later stages of inflammation.

## 5. Conclusions and Perspectives

The last four decades have witnessed remarkable progress in our understanding of how immune cells traverse vast distances between tissues and secondary lymphoid organs to engage the innate and adaptive immune systems and mount an effective immune response. More recently, neutrophil lymphatic migration has been shown to be an important contributor to this process. This led to a paradigm shift in our view of neutrophils: from short-lived effectors designed to provide sterilizing immunity at sites of infection, to multifunctional immune cells that activate adaptive immunity in draining lymph nodes [120,121]. The rapid entry of neutrophils into lymphatic vessels and their subsequent transmigration to lymph nodes places these leukocytes in prime position to provide early signals to adaptive immune cells [9,10,11,12].

Neutrophils entering lymph nodes via the two different routes (HEVs versus afferent lymphatics) appear to have distinct phenotypes and functions. The majority of neutrophils arriving in draining lymph nodes are recruited from the circulation in response to inflammation in the lymph node and play an important role in preventing further pathogen dissemination [15,122]. Neutrophils entering through the afferent lymphatics make up a smaller proportion of lymph node neutrophils but have an activated APC-like phenotype and often transport antigens from infected tissues [9,10], suggesting that they may have a more prominent role in interacting with lymph node immune cells and modulating adaptive immunity. The signals that direct neutrophils away from sites of inflammation in the periphery into lymphatic vessels and draining lymph nodes are yet to be fully understood. The two subpopulations of lymph node neutrophils, i.e., entering the lymph node from the circulation or inflamed tissues, may provide a possible explanation for the diverse effects of neutrophil depletion on the adaptive immune response [123,124].

Inhibiting leukocyte trafficking is an effective strategy for treating a myriad of inflammatory diseases [125,126]. Since tissue-egressing neutrophils and circulating neutrophils employ distinct molecular mechanisms and different routes of lymph node entry, it may be possible to specifically inhibit recruitment of one specific subpopulation. This suggests that treatment can be tailored to block one neutrophil subset but not the other and modulate a specific aspect of neutrophil interactions with the lymphatic system.

A less understood facet of neutrophil interactions with the lymphatic system is their ability to promote inflammatory lymphangiogenesis. Lymphangiogeneis has been associated with transplant rejection [127,128] and inhibiting this process improves graft survival in corneal transplantation [105]. However, the induction of therapeutic lymphangiogenesis mitigates allograft rejection in a orthotopic model of lung transplantation [129]. In addition, preclinical studies in models of inflammatory skin disease [117,130] demonstrated that promoting lymphangiogenesis may contribute to the resolution of inflammation. This suggests that neutrophil-driven stimulation of lymphangiogenesis may have two distinct and context-dependent roles in inflammation, i.e., promoting pathological inflammation or contributing to its resolution. The discovery of neutrophil-derived vascular endothelial factors involved in lymphangiogenesis presents an opportunity to develop targeted therapies for inflammatory disorders once these opposing roles are fully understood. Thus, clarifying neutrophil interactions with the lymphatic system is a crucial aspect in developing neutrophil-targeted interventions in inflammatory diseases.

## Figures and Tables

**Figure 1 cells-10-02106-f001:**
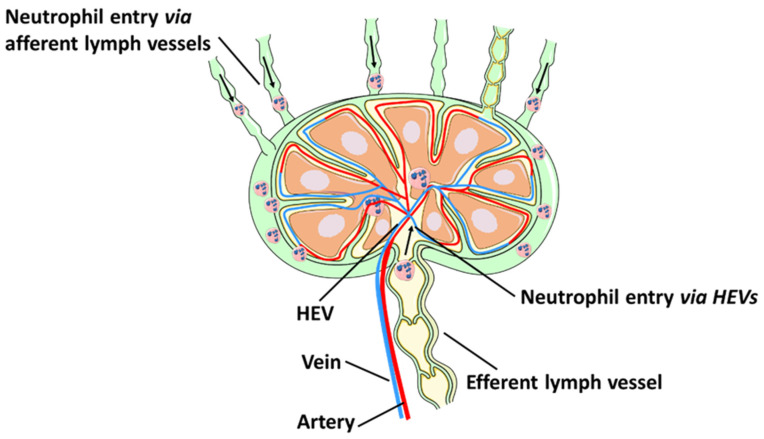
Neutrophil migration to lymph nodes. Neutrophils rapidly migrate from the site of inflammation into lymphatic vessels and draining lymph nodes. Neutrophils can also enter the lymph nodes from circulation by crossing HEVs.

**Figure 2 cells-10-02106-f002:**
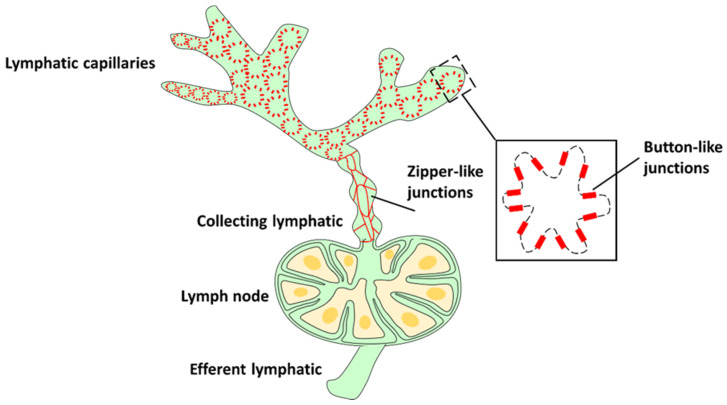
Organization of the lymphatic network. Afferent lymphatics start as blind-ended capillaries in peripheral tissues [32]. These then branch and merge with collecting vessels, which feed into a draining lymph node [33]. The endothelium of collecting vessels has continuous zippers that prevent vascular leaks and allow transport of the lymph. Lymph fluid and leukocytes leave the lymph node through the efferent collecting vessels.

**Figure 3 cells-10-02106-f003:**
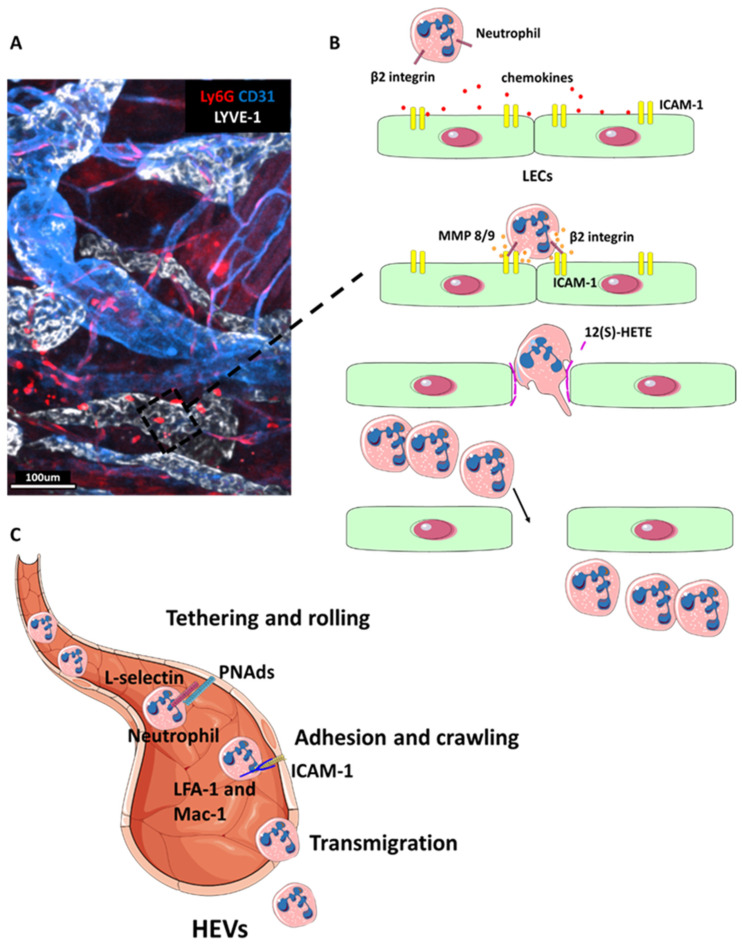
Neutrophil migration to lymphoid organs. (**A**) Two-photon microscopy image of a section of inflamed skin in a Ly6G-Tomato reporter mouse [48] stained with LYVE-1 (white) to visualise LECs and endothelial cells with CD31 (blue). Neutrophils (red) can be seen in and around lymphatic vessels. (**B**) Schematic representation of neutrophil transmigration across lymphatic endothelium. First, neutrophils chemotax toward LECs following inflammation-induced release of chemokines such as CXCL8 in humans and CXCL1 in mice [49]. Next, neutrophils adhere to inflamed LECs via integrin binding to ICAM-1. This triggers the secretion of MMP8, MMP9 and exocytosis of chemorepellent 12(S)-HETE, which together promote transient junctional retraction and neutrophil transmigration. The resulting gap in the lymphatic endothelium serves as a hotspot for rapid neutrophil transmigration. Adapted from D.G. Jackson, *Frontier Immunology*, 2019 [2]. (**C**) Diagram showing neutrophil entry into the lymph node via HEVs. Neutrophil express L-selectin which binds to glycoproteins expressed on the HEVs walls, such as peripheral node addressins (PNAds) to allow tethering and rolling, which promote rapid adhesion and emigration [15]. Neutrophil adhesion is mediated via binding of ICAM-1 to integrins LFA-1 and Mac-1. In particular, LFA-1 mediates neutrophil adhesion and Mac-1 facilitates subsequent crawling [50].

**Figure 4 cells-10-02106-f004:**
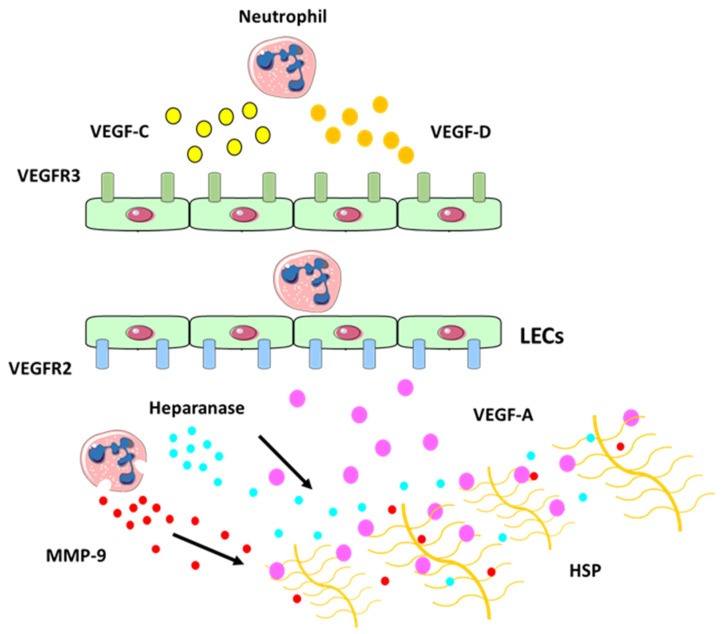
Neutrophil contribution to inflammatory lymphangiogenesis. Schematic illustrating neutrophil contributions to lymphangiogenesis at the site of inflammation. Neutrophils can secrete VEGF-C (in yellow) and VEGF-D (in orange) at the site of inflammation which bind to their receptor VEGFR-3 (in green) expressed on LECs. Neutrophils can also increase VEGF-A (in purple) availability by secreting MMP-9 and heparanase which cleaves heparan sulphate preoteoglycans (HSP) side chains and release the HSP-trapped VEGF-A. This binds to VEGFR-2 receptor (in blue), which promotes the expansion of the lymphatic vessel network [119].

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
