# Peer review of "Neutrophil Interactions with the Lymphatic System"

_cells, 2021, doi:10.3390/cells10082106_

Round 1

Reviewer 1 Report

The review by Jakovija and Chtanova is a well written and interesting perspective on the role of neutrophils in lymph nodes and the routes that they take to get there. Comments below are provided to help with clarity and flow.

  1. Definition of terms would greatly help with clarity, especially for a naive reviewer. The authors use migration in the lymphatic system to mean migration into the lymph node either via a lymphatic or hematogenous route. I found this confusing because they also talk about lymphatic migration to mean via lymphatic vessels. 
  2. In introducing the routes of neutrophil migration to lymph nodes, either through HEV or via afferent lymphatic vessels, the authors move between the  two almost as though they are equivalent. Particularly in the last paragraph on page 1 (line 40) the first sentence says neutrophils go to LNs, the second that you find them in lymph, and the third that they get there in a lot of cases through blood...It's not clear when a lymphatic route is used vs hematogenous or whether this matters. It's not until the conclusion that the authors begin to speculate on the idea that there could be different neutrophil phenotypes taking these different routes that might lead to unique function. It would be helpful to discuss this up front.
  3. Similarly, it seems that the authors are suggesting that neutrophils always egress from peripheral tissue to LNs via afferent lymphatic vessels but the data to directly support this is fairly limited. A more specific discussion of the context dependence is necessary - or a statement that it needs to be investigated more to understand if a common feature of inflamed tissue.
  4. The authors book end their discussion of the role of neutrophils in the LN with a description of mechanisms that regulatelymphangiogenesis and then how neutrophils might promote lymphangiogenesis at the end. Its not really clear why. It might be better to merge these paragraphs together to illustrate the fact that lymphatic vessels (and presumably blood as well) are not only routes for neutrophil migration but also targets. 
  5. If discussing neutrophils and lymphangiogenesis, a discussion of why lymphangiogenesis matters seems warranted. As written it is assumed that the reader knows why this might be important and would feedback on lymph node immune responses.
  6. The figures are a bit sparse and very lymphatic vessel centric despite the fact that a lot of the review is generally about neutrophil migration into lymph nodes from blood as well. Can the authors provide some figures that better integrate the ideas from the review? Doesn't seem really necessary to include schematic of buttons vs zippers, as these are not well discussed in the text.

Minor comments:

Line 68 needs references. Though also not quite sure why its included.

Lines 133-137, it is not clear what is meant by neutrophils in branches. Bifurcations of what?

Line 233, statement is that neutrophils are the most abundant subset migrating to LNs via lymphatics at early time points, but only one reference and only one context investigated. This is a pretty broad statement given that.

Author Response

The review by Jakovija and Chtanova is a well written and interesting perspective on the role of neutrophils in lymph nodes and the routes that they take to get there. Comments below are provided to help with clarity and flow.

Thank you for a very thorough review of our manuscript and thoughtful suggestions. We have revised our review as suggested. Specific changes are detailed below. 

Definition of terms would greatly help with clarity, especially for a naive reviewer. The authors use migration in the lymphatic system to mean migration into the lymph node either via a lymphatic or hematogenous route. I found this confusing because they also talk about lymphatic migration to mean via lymphatic vessels. 

Thank you for pointing this out. As suggested, we clarified the terminology in our discussion of neutrophil migration. We use “lymphatic migration” to mean migration via lymphatic vessels and avoid the term “migration in the lymphatic system”.

In introducing the routes of neutrophil migration to lymph nodes, either through HEV or via afferent lymphatic vessels, the authors move between the two almost as though they are equivalent. Particularly in the last paragraph on page 1 (line 40) the first sentence says neutrophils go to LNs, the second that you find them in lymph, and the third that they get there in a lot of cases through blood...It's not clear when a lymphatic route is used vs hematogenous or whether this matters. It's not until the conclusion that the authors begin to speculate on the idea that there could be different neutrophil phenotypes taking these different routes that might lead to unique function. It would be helpful to discuss this up front.

As suggested, we have revised our introduction to bring in the concept of distinct neutrophil phenotypes in lymph nodes reflecting their distinct origin and why distinguishing different routes of entry is important.

“Lymph node neutrophils are phenotypically distinct from circulating neutrophils and have the capacity to modulate adaptive immunity (Lok, Dennison et al. 2019). Although the existence of bona fide neutrophils subsets remains to be conclusively demonstrated, the two different routes of neutrophil entry (via afferent lymphatics or HEVs) suggest that the two populations of lymph node neutrophils may have distinct functions. Neutrophils entering the lymph node via afferent lymphatics are arriving from sites of inflammation, are highly activated, express costimulatory molecules and carry tissue antigens (Abadie, Badell et al. 2005, Hampton, Bailey et al. 2015). This may enable them to act as antigen presenting cells as well as shuttle pathogens to draining lymph nodes. On the other hand, neutrophils arriving from circulation are likely coming from the bone marrow (although it is possible that a proportion of tissue neutrophils undergoes reverse transmigration to enter circulation  (Mathias, Perrin et al. 2006, Wang, Hossain et al. 2017) and then migrates to draining lymph nodes) in response to pathogens in the lymph nodes. It is likely that neutrophils entering from circulation may play a prominent role in preventing systemic dissemination of bacteria (Bogoslowski, Butcher et al. 2018).”

Similarly, it seems that the authors are suggesting that neutrophils always egress from peripheral tissue to LNs via afferent lymphatic vessels but the data to directly support this is fairly limited. A more specific discussion of the context dependence is necessary - or a statement that it needs to be investigated more to understand if a common feature of inflamed tissue.

We revised our manuscript to include a discussion of the possible fates for tissue neutrophils including death in situ, reverse transmigration into blood and egress into lymphatics. We also added a section discussing when lymphatic egress occurs.

The authors book end their discussion of the role of neutrophils in the LN with a description of mechanisms that regulate lymphangiogenesis and then how neutrophils might promote lymphangiogenesis at the end. Its not really clear why. It might be better to merge these paragraphs together to illustrate the fact that lymphatic vessels (and presumably blood as well) are not only routes for neutrophil migration but also targets. 

Thank you for the suggestion. We merged the two sections on lymphangiogenesis and moved the new subsection to the section on neutrophil functions within the lymphatic system.

If discussing neutrophils and lymphangiogenesis, a discussion of why lymphangiogenesis matters seems warranted. As written it is assumed that the reader knows why this might be important and would feedback on lymph node immune responses.

As suggested, we added a section summarising what is currently known about the possible effects of lymphangiogenesis on the immune response:

“The functional consequences of inflammation-induced lymphangiogenesis require further investigation. In some settings, expansion of the lymphatic network may support inflammation by promoting transport of immune cells and antigens to the lymph node and enhancing the immune response. In other settings,  lymphatic vessel expansion contributes to resolution of inflammation by draining inflammatory mediators and cells from the site of inflammation (Schwager and Detmar 2019). For instance, inhibiting the lymphatic vasculature led to increased inflammation in mouse models of skin inflammation (Kajiya and Detmar 2006, Kataru, Jung et al. 2009, Huggenberger, Ullmann et al. 2010), inflammatory bowel disease (Jurisic, Sundberg et al. 2013, D'Alessio, Correale et al. 2014) and rheumatoid arthritis (Guo, Zhou et al. 2009). However, inhibiting this process improves graft survival in corneal transplantation (Amouzegar, Chauhan et al. 2016, Hou, Bock et al. 2021).”

The figures are a bit sparse and very lymphatic vessel centric despite the fact that a lot of the review is generally about neutrophil migration into lymph nodes from blood as well. Can the authors provide some figures that better integrate the ideas from the review? Doesn't seem really necessary to include schematic of buttons vs zippers, as these are not well discussed in the text.

We added an extra figure (new Figure 1) showing neutrophil entry into lymph nodes via afferent lymphatics and HEVs. We have also expanded Figure 3 to incorporate migration via HEVs. New Figure 2 displaying the schematic of lymphatic vasculature was modified in line with suggestions from the other reviewers and manuscript text was adjusted to better integrate this figure in the review.

Minor comments:

Line 68 needs references. Though also not quite sure why its included.

 We changed the line to read:

“Moreover, lymphatic vessels are important for macrophage clearance during the resolution phase of tissue inflammation and infections (Jackson 2019) and for maintenance of peripheral immune tolerance (Liao and von der Weid 2014). “

Lines 133-137, it is not clear what is meant by neutrophils in branches. Bifurcations of what?

We have clarified our manuscript to specify bifurcations of capillaries.

Line 233, statement is that neutrophils are the most abundant subset migrating to LNs via lymphatics at early time points, but only one reference and only one context investigated. This is a pretty broad statement given that.

We have changed the text to read:

“Neutrophils can ferry antigens from the site of infection to the draining lymph node. At the early timepoints they are the first immune subset migrating to draining lymph nodes (Abadie, Badell et al. 2005, Chtanova, Schaeffer et al. 2008, Lämmermann, Afonso et al. 2013, Hampton, Bailey et al. 2015). This suggests that neutrophils are important contributors to the regulation of early adaptive immune responses against foreign antigens or infectious agents, following their rapid migration into the lymphatic system (Abadie, Badell et al. 2005, Neeland, Shi et al. 2016).”

Reviewer 2 Report

In this review, the authors discuss how neutrophils (PMNs) enter and travel within the lymphatic vasculature. They highlight the multifaceted roles of PMNs in the innate and adaptive immune responses. They discuss the dual role of PMNs as pathogen killers and as Trojans horses for a few intracellular pathogens.

Comments: The review focus on PMN migration into lymphoid organs and trafficking within the lymphatic vasculature. There is increasing evidence for subpopulations of neutrophils. Of course, this point is stress in the concluding remarks, but providing more details on the phenotypes of PMNs (if known) that enter and circulate within the lymphatic vasculature would strengthen the review.  

Minor point:

Page 4, line 128: What HEV stand for?  If it is for high endothelial venule, please define the abbreviation on line 123 and use “HEV” consistently throughout the manuscript.

Author Response

In this review, the authors discuss how neutrophils (PMNs) enter and travel within the lymphatic vasculature. They highlight the multifaceted roles of PMNs in the innate and adaptive immune responses. They discuss the dual role of PMNs as pathogen killers and as Trojans horses for a few intracellular pathogens.

Comments: The review focus on PMN migration into lymphoid organs and trafficking within the lymphatic vasculature. There is increasing evidence for subpopulations of neutrophils. Of course, this point is stress in the concluding remarks, but providing more details on the phenotypes of PMNs (if known) that enter and circulate within the lymphatic vasculature would strengthen the review.  

Thank you for your review of our manuscript and helpful suggestions. We have revised our manuscript according to the suggestion to provide more details about neutrophil subpopulations in the introduction (Lines 57-70 ). In addition, the discussion of the different phenotypes is expanded in the section on neutrophil migration to lymphoid organs (Lines 165-190/202-222 ).

Minor point:

Page 4, line 128: What HEV stand for?  If it is for high endothelial venule, please define the abbreviation on line 123 and use “HEV” consistently throughout the manuscript.

We apologise for the oversight. We have now defined abbreviations at first use and used consistently.

Reviewer 3 Report

The review by Jakovija and Chtanova summarizes our current knowledge of interactions occurring between neutrophils and the lymphatic system. This includes physical interactions of neutrophils with endothelial cells, i.e. when trafficking via afferent lymphatics or across high endothelial venules, but also their presumed immune functions in lymph nodes as well as their contribution to lymphangiogenesis. The range of topics covered is therefore quite broad, and it is nice to see everything compiled in one review. While the literature is typically accurately summarized in the individual review sections, the review as a whole would benefit from better transitions between the sections and more conclusions and discussion of the accumulated findings.

Here are a few suggestions for improvement:

  1. The topic of the Review is about neutrophil interactions with the lymphatic system, what includes leukocyte entry through HEVs and the general actions of neutrophils in lymph nodes. In fact, most of these aspects are covered within this review. This is in contrast to the last sentence of the abstract, which restricts the review content to neutrophil trafficking within the lymphatic vasculature. – The last sentence should therefore be corrected to reflect the true content of the review.

  1. Role of neutrophils in induction of lymphangiogenesis in draining lymph nodes and in the inflamed tissue: In the abstract, both sites (lymph node and tissue) are mentioned, but section 5 (line 307-337) only deals with the site of inflammation, not the lymph node. Considering that there is not so much literature on this topic altogether, I would recommend to also include a discussion of neutrophil-induced LN lymphangiogenesis.

  1. Paragraph on lymphatic vasculature (lines 57-82): when talking about the general immune and transport functions of the lymphatic vasculature, it would be relevant to also mention the emerging functions of LECs as antigen-presenting cells. Moreover, while the paragraph mentions the junctional structure of lymphatic capillaries (line 77/78), it does not yet explain the setup in collecting vessels. Finally, references 21 – 23 are not found in the paragraph (jump from 20 to 24).

  1. General comment on Figures: please write the labels in a bigger font size; there typically are lots of (empty) spaces, which seems not optimally used.

  1. Paragraph on neutrophil migration to lymphoid organs (lines 110 onwards):

The first sentence states that neutrophils have been detected in afferent lymph in sheep and in murine lymph nodes. Please be more complete and also mention studies on e.g. human afferent lymph and human lymph nodes and on the general composition of lymph.

When mentioning the IVM studies performed (only a handful so far, to my knowledge), please also cite Arokiasamy et al., Sci. Reports 2017 (neutrophils in cremaster lymphatics).

  1. Line 152: “In contrast to migration though high endothelial venules, neutrophil entry form inflamed tissues into afferent lymphatics involves b2 integrins”. To my knowledge, the authors themselves showed that entry via HEVs into LNs is integrin b2-dependent (Hampton et al., Nat. Comm. 2015). Thus, please correct this sentence.

  1. Paragraph on chemokine-dependence of neutrophil migration into afferent lymphatics (lines 171-183): Please also add the Arokiasamy et al., Sci Rep. 2017 as another example of CCR7-dependent migration into lymphatic vessels. If the authors think that the chemokine/receptor requirement is stimulus-dependent, it might be good to state this more clearly (i.e. to bring the different findings in perspective). Line 178: Please provide a more appropriate reference for the expression of the different CCR7 ligands (e.g. a review). Lines 184-186: This sentence seems out of place here (in the chemokine section) and should rather be moved further up to where adhesion molecules are discussed.

  1. Section 3.1. (Molecular mechanisms of neutrophil lymphatic migration) is so far not mentioning anything about the arrival of lymph-borne neutrophils in the subcapsular or medullary sinus and translocation into the LN parenchyma. Even though little might be known about this, it would be important to still mention it and cite a recent paper from the group of R. Förster (https://doi.org/10.3390/cells10061486) and further studies, if there are any.

  1. Figure Legend of Figure 2: Please provide a scale bar for the confocal image shown in 2A and write the labels (colors) directly into the image. Lines 193/194: what is meant by inflammation-induced chemokine release? Which chemokines are the authors referring to?

  1. Section 4: Neutrophil functions within the lymphatic system (lines 201 – 306). Considering that the main focus of this Special Issue series is on lymphatic migration and interactions with lymphatic endothelium, this section appears a bit long. In particular, it would be better if the relevance of lymphatic migration for the execution of the different functions was better detailed. Actually, this only became clearer to me after reading the “conclusion and perspectives” section (lines 353-356). Perhaps this concept / hypothesis could already be more accentuated in section 4.

  1. Section 5. Neutrophils as important contributors to lymphangiogenesis:

As mentioned before, this section should be expanded and also include the impact of neutrophils on LN lymphangiogenesis.

The authors mention the role of MMP-9 in enhancing the bioavailability of VEGF-A (lines 313-315): please cite the corresponding primary literature at the end of the sentence (I believe Tan et al., Blood 2013) and explain in the text how this works mechanistically. This is necessary, since the concept is also highlighted in Figure 3, but not sufficiently explained at present.

Line 316: Please find a more suitable reference for this statement (perhaps a review article).

Lines 316-318: It is not clear to me why this sentence is mentioned, considering that the reference does not talk about neutrophils nor about lymphangiogenesis. Moreover, VEGF-A exists in a soluble and ECM bound form (VEGF-A-121 & 165); if the connection to HSP is made, this would need to be better explained.

Figure 3: In its present state it is not very informative and perhaps even a bit misleading: Why are VEGFR2 and VEGFR-3 only expressed in the central parts of the vessel? Why is the vessel wider in the upper part? What is the MMP9 doing? Again, please explain the mechanism.

Line 337: please provide a more general reference (review or several references) for VEGF-A/VEGFR2-mediated expansion of the lymphatic network (Ref. 92 is a developmental study).

  1. Section 6. Conclusions and perspectives: Role of lymphangiogenesis in transplant rejection (line 369-372): ..”inhibiting this process improves graft survival”(line 370). While this statement is true for the cornea (i.e. the reference provided), the role of lymphangiogenesis is not so clear in other organs (see e.g. PMID: 26485284). Please make sure that the text reflects this ambiguity.

  1. General comment on references: if you are citing more than one study, please fuse the references (should be within one bracket). In the references section: please provide complete author lists (not …et al.), including the final author.

  1. General comment on abbreviations: introduce once and use consistently (e.g. high endothelial venules, MMP-9)

  1. Several reviews that are cited are very old and should be replaced (or accompanied) by more up-to-date references. E.g. references 28 (line 96), 56 (line 213), 95 (line 364).

Author Response

The review by Jakovija and Chtanova summarizes our current knowledge of interactions occurring between neutrophils and the lymphatic system. This includes physical interactions of neutrophils with endothelial cells, i.e. when trafficking via afferent lymphatics or across high endothelial venules, but also their presumed immune functions in lymph nodes as well as their contribution to lymphangiogenesis. The range of topics covered is therefore quite broad, and it is nice to see everything compiled in one review. While the literature is typically accurately summarized in the individual review sections, the review as a whole would benefit from better transitions between the sections and more conclusions and discussion of the accumulated findings.

 We thank the reviewer for a thoughtful and thorough review of our manuscript. We made extensive revisions as suggested and hope that the revised manuscript is significantly improved.

Here are a few suggestions for improvement:

  1. The topic of the Review is about neutrophil interactions with the lymphatic system, what includes leukocyte entry through HEVs and the general actions of neutrophils in lymph nodes. In fact, most of these aspects are covered within this review. This is in contrast to the last sentence of the abstract, which restricts the review content to neutrophil trafficking within the lymphatic vasculature. – The last sentence should therefore be corrected to reflect the true content of the review.

Thank you for pointing this out. We revised the abstract as suggested to read:

“In this review, we discuss the significance of neutrophil migration to secondary lymphoid organs and within the lymphatic vasculature and highlight emerging evidence of the neutrophil’s role in lymphangiogenesis.”

  1. Role of neutrophils in induction of lymphangiogenesis in draining lymph nodes and in the inflamed tissue: In the abstract, both sites (lymph node and tissue) are mentioned, but section 5 (line 307-337) only deals with the site of inflammation, not the lymph node. Considering that there is not so much literature on this topic altogether, I would recommend to also include a discussion of neutrophil-induced LN lymphangiogenesis.

We revised as suggested to include neutrophil-induced lymphangiogenesis:

“Similarly to tissue lymphangiogenesis, VEGF-A-VEGFR2 and VEGF-C/D-VEGFR3 are the principal mediators of lymph node (or intranodal) lymphangiogenesis (Ji 2009). During inflammation, B cells (Shrestha, Hashiguchi et al. 2010), macrophages recruited from peripheral tissues (Kataru, Jung et al. 2009) and fibroblast-type reticular stromal cells (Chyou, Ekland et al. 2008) are the major sources of VEGF-A in the lymph node. Emerging evidence suggests that neutrophils also contribute to lymph node lymphangiogenesis (Tan, Chong et al. 2013). In mice lacking B cells, neutrophils compensate to support lymph lymphangiogenesis during prolonged inflammation (Tan, Chong et al. 2013). However, more studies are needed to determine whether specific neutrophil subsets are involved in this process.”

Paragraph on lymphatic vasculature (lines 57-82): when talking about the general immune and transport functions of the lymphatic vasculature, it would be relevant to also mention the emerging functions of LECs as antigen-presenting cells. Moreover, while the paragraph mentions the junctional structure of lymphatic capillaries (line 77/78), it does not yet explain the setup in collecting vessels. Finally, references 21 – 23 are not found in the paragraph (jump from 20 to 24).

Thank you for the suggestions. We have revised as follows:

“Lymphatic endothelial cells (LECs) are exposed to peripheral extracellular antigens after tissue damage, infection or inflammation (Hirosue, Vokali et al. 2014) and can take up these antigens and present them to T cells (Vokali, Yu et al. 2020, Gkountidi, Garnier et al. 2021)”.

“Collecting lymphatic vessels are uniquely adapted to ensure leakage-free transport of lymph (Schineis, Runge et al. 2019). They contain valves which ensure unidirectional flow of lymph and prevent backflow (Bazigou, Xie et al. 2009). Collecting LECs have an elongated morphology, are connected by continuous and tight zipper-like cell-cell junctions (Baluk, Fuxe et al. 2007) and enclosed by a continuous basement membrane covered by smooth muscle cells (Zhang, Zarkada et al. 2020).”

We have also fixed references.

  1. General comment on Figures: please write the labels in a bigger font size; there typically are lots of (empty) spaces, which seems not optimally used.

We increased font sizes as suggested.

  1. Paragraph on neutrophil migration to lymphoid organs (lines 110 onwards):

The first sentence states that neutrophils have been detected in afferent lymph in sheep and in murine lymph nodes. Please be more complete and also mention studies on e.g. human afferent lymph and human lymph nodes and on the general composition of lymph.

We have added a section on the general composition of lymph.

“The lymph is the biological fluid, whose function is to return the interstitial fluid from tissues to the blood circulation (Santambrogio 2018). Since lymph is produced as the result of a capillary  filtration process, it’s composition is very similar to the plasma ultrafiltrate with high numbers of lymphocytes, macrophages and plasma proteins (Dzieciatkowska, D’Alessandro et al. 2014). In addition, lymph contains a large amount of dietary fats, related lipoproteins and cholesterol absorbed from the intestine (Meng and Veenstra 2007).”

We provided more detail about afferent lymph in sheep and murine lymph nodes and added studies in humans.

“Early cannulation studies of afferent lymphatic vessels conducted in sheep and humans under homeostatic, conditions detected low number of neutrophils in the afferent lymph (Hall and Morris 1962, Smith, McIntosh et al. 1970, Olszewski 1995, EGAN, KIMPTON et al. 1996, HAIG, HOPKINS et al. 1999). Neutrophil numbers increased dramatically but transiently in sheep afferent lymph following vaccination with liposomes containing an innate cell stimulator poly(I:C) and diptheria toxoid (DT) (de Veer, Neeland et al. 2013). These studies provided some of the earliest evidence for neutrophil migration in lymphatic vessels.”

“Recently, in vivo labelling (Abadie, Badell et al. 2005, Hampton, Bailey et al. 2015) and advances in intravital imaging technologies  (Arokiasamy, Zakian et al. 2017, Yam and Chtanova 2020) allowed tracking of neutrophils migrating from the periphery to draining lymph nodes and visualisation of their interactions with the lymphatic vessels in mice (Hampton, Bailey et al. 2015, Arokiasamy, Zakian et al. 2017). “

When mentioning the IVM studies performed (only a handful so far, to my knowledge), please also cite Arokiasamy et al., Sci. Reports 2017 (neutrophils in cremaster lymphatics).

We apologise for the oversight. We have now included this study in the review (Reference 12).

“Recently, in vivo labelling (Abadie, Badell et al. 2005, Hampton, Bailey et al. 2015) and advances in intravital imaging technologies  (Arokiasamy, Zakian et al. 2017, Yam and Chtanova 2020) allowed tracking of neutrophils migrating from the periphery to draining lymph nodes and visualisation of their interactions with the lymphatic vessels in mice (Hampton, Bailey et al. 2015, Arokiasamy, Zakian et al. 2017). In a mouse model of cremaster muscle inflammation, antigen sensitisation led to a rapid transient entry of tissue neutrophils into lymphatic vessels and subsequent crawling along the luminal side of the lymphatic endothelium (Arokiasamy, Zakian et al. 2017).”

  1. Line 152: “In contrast to migration though high endothelial venules, neutrophil entry form inflamed tissues into afferent lymphatics involves b2 integrins”. To my knowledge, the authors themselves showed that entry via HEVs into LNs is integrin b2-dependent (Hampton et al., Nat. Comm. 2015). Thus, please correct this sentence.

We revised this section as follows:

“Neutrophil entry from inflamed tissues into afferent lymphatics involves β2 integrins (Rigby, Ferguson et al. 2015). Indeed, adhesion molecules, such as ICAM-1 and VCAM-1 (expressed on LECs), can support human and mouse neutrophil lymphatic interactions via binding to leucocyte β2 integrin (Rigby, Ferguson et al. 2015). Integrin‐mediated neutrophil adhesion to inflamed LECs through ICAM-1 binding induces the release of matrix metalloproteinases MMP8 and MMP9 and secretion of 12-hydroxyeicosatetraenoate (12(S) HETE)(Jackson 2019).  Neutrophil migration in lymphatic vessels also requires CD11b (or integrin alpha M, which binds to integrin β2 to form complement receptor 3 (Solovjov, Pluskota et al. 2005)), since inhibition or complete removal of CD11b inhibited neutrophil migration from the skin to draining lymph nodes (Hampton, Bailey et al. 2015). “

  1. Paragraph on chemokine-dependence of neutrophil migration into afferent lymphatics (lines 171-183): Please also add the Arokiasamy et al., Sci Rep. 2017 as another example of CCR7-dependent migration into lymphatic vessels. If the authors think that the chemokine/receptor requirement is stimulus-dependent, it might be good to state this more clearly (i.e. to bring the different findings in perspective). Line 178: Please provide a more appropriate reference for the expression of the different CCR7 ligands (e.g. a review). Lines 184-186: This sentence seems out of place here (in the chemokine section) and should rather be moved further up to where adhesion molecules are discussed.

We revised this section as suggested to read:

“Neutrophil migration to draining lymph nodes also appears to rely on CCR7 in some settings (Beauvillain, Cunin et al. 2011). For instance, in models of TNF-α- and/or Complete Freund's adjuvant (CFA)-induced inflammation in the skin (Beauvillain, Cunin et al. 2011) or cremaster muscle (Arokiasamy, Zakian et al. 2017) neutrophil migration to lymph nodes was CCR7-dependent. Neutrophils upregulated CCR7 upon entry into inflamed tissues suggesting that neutrophil priming is required for enhanced chemokine receptor expression (Arokiasamy, Zakian et al. 2017). On the other hand, neutrophil egress from the skin and entry into lymph nodes was independent of CCR7 after S. aureus administration to the skin (Hampton, Bailey et al. 2015). These opposing outcomes suggest that the requirement for CCR7 is stimulus dependent and may be influenced by neutrophil phenotype, activation status, tissue microenvironment and other factors. This also indicates that additional chemokine pathways can mediate neutrophil lymphatic migration (Hampton, Bailey et al. 2015).”

  1. Section 3.1. (Molecular mechanisms of neutrophil lymphatic migration) is so far not mentioning anything about the arrival of lymph-borne neutrophils in the subcapsular or medullary sinus and translocation into the LN parenchyma. Even though little might be known about this, it would be important to still mention it and cite a recent paper from the group of R. Förster (https://doi.org/10.3390/cells10061486) and further studies, if there are any.

We apologise for the oversight. We have revised this section to include the above study.

“Neutrophil localization in lymph nodes may reflect their role in pathogen containment and immune cell interactions. For instance, adoptive transfer of neutrophils into afferent lymphatic vessels showed that lymph derived neutrophils localized in the subcapsular or medullary sinus of the lymph node (de Castro Pinho and Förster 2021). “

  1. Figure Legend of Figure 2: Please provide a scale bar for the confocal image shown in 2A and write the labels (colors) directly into the image. Lines 193/194: what is meant by inflammation-induced chemokine release? Which chemokines are the authors referring to?

We re-labelled the image and clarified chemokine identity.

“First, neutrophils chemotax toward LECs following inflammation-induced release of chemokines such as CXCL8 in humans and CXCL1 in mice (Vigl, Aebischer et al. 2011).”

  1. Section 4: Neutrophil functions within the lymphatic system (lines 201 – 306). Considering that the main focus of this Special Issue series is on lymphatic migration and interactions with lymphatic endothelium, this section appears a bit long. In particular, it would be better if the relevance of lymphatic migration for the execution of the different functions was better detailed. Actually, this only became clearer to me after reading the “conclusion and perspectives” section (lines 353-356). Perhaps this concept / hypothesis could already be more accentuated in section 4.

We shortened this section and highlighted the hypothesis of different neutrophil functional subsets.

Section 5. Neutrophils as important contributors to lymphangiogenesis:

As mentioned before, this section should be expanded and also include the impact of neutrophils on LN lymphangiogenesis.

We have revised this section as suggested to include lymphangiogenesis.

The authors mention the role of MMP-9 in enhancing the bioavailability of VEGF-A (lines 313-315): please cite the corresponding primary literature at the end of the sentence (I believe Tan et al., Blood 2013) and explain in the text how this works mechanistically. This is necessary, since the concept is also highlighted in Figure 3, but not sufficiently explained at present.

We have revised this section as suggested:

“Neutrophils secrete VEGF-D at sites of inflammation, which promotes proliferation of LECs, stimulates lymphangiogenesis (Veikkola, Jussila et al. 2001) and plays an important role during the development of the lymphatic system (Baldwin, Halford et al. 2005). Neutrophils may also increase VEGF-A bioavailability (Tan, Chong et al. 2013). In fact, after secretion VEGF becomes bound to the ECM and the interaction between VEGF-A and matrix proteins is mediated through the carboxy terminal region, also known as heparin-binding or ECM-binding domain (Houck, Leung et al. 1992). This domain is able to bind heparin sulfate proteoglycans and other matrix proteins (Houck, Leung et al. 1992, Poltorak, Cohen et al. 1997) and is thought to be responsible for the sequestration of VEGF-A within the matrix. Neutrophils act via MMP-9 and heparanase (which cleaves heparan sulfate preoteoglycans (HSP) side chains and releases the HSP-trapped VEGF-A) (Figure 4) to increase the amount of biologically active VEGF-A which in turn activated inflammatory lymphangiogenesis (Tan, Chong et al. 2013). “

Line 316: Please find a more suitable reference for this statement (perhaps a review article).

We changed the paragraph as detailed above.  

Lines 316-318: It is not clear to me why this sentence is mentioned, considering that the reference does not talk about neutrophils nor about lymphangiogenesis. Moreover, VEGF-A exists in a soluble and ECM bound form (VEGF-A-121 & 165); if the connection to HSP is made, this would need to be better explained.

Thank you for the suggestion. We made the following revisions:

“Neutrophils secrete VEGF-D at sites of inflammation, which promotes proliferation of LECs, stimulates lymphangiogenesis (Veikkola, Jussila et al. 2001) and plays an important role during the development of the lymphatic system (Baldwin, Halford et al. 2005). Neutrophils may also increase VEGF-A bioavailability (Tan, Chong et al. 2013). In fact, after secretion VEGF becomes bound to the ECM and the interaction between VEGF-A and matrix proteins is mediated through the carboxy terminal region, also known as heparin-binding or ECM-binding domain (Houck, Leung et al. 1992). This domain is able to bind heparin sulfate proteoglycans and other matrix proteins (Houck, Leung et al. 1992, Poltorak, Cohen et al. 1997) and is thought to be responsible for the sequestration of VEGF-A within the matrix. Neutrophils act via MMP-9 and heparanase (which cleaves heparan sulfate preoteoglycans (HSP) side chains and releases the HSP-trapped VEGF-A) (Figure 4) to increase the amount of biologically active VEGF-A which in turn activated inflammatory lymphangiogenesis (Tan, Chong et al. 2013).”

Figure 3: In its present state it is not very informative and perhaps even a bit misleading: Why are VEGFR2 and VEGFR-3 only expressed in the central parts of the vessel? Why is the vessel wider in the upper part? What is the MMP9 doing? Again, please explain the mechanism.

We revised the figure as suggested.

Line 337: please provide a more general reference (review or several references) for VEGF-A/VEGFR2-mediated expansion of the lymphatic network (Ref. 92 is a developmental study).

“Neutrophils can also increase VEGF-A (in purple) availability by secreting MMP-9 and heparanase which cleaves heparan sulfate preoteoglycans (HSP) side chains and release the HSP-trapped VEGF-A. This binds to VEGFR-2 receptor (in blue), which promotes the expansion of the lymphatic vessel network (Koch and Claesson-Welsh 2012). “

  1. Section 6. Conclusions and perspectives: Role of lymphangiogenesis in transplant rejection (line 369-372): ..”inhibiting this process improves graft survival”(line 370). While this statement is true for the cornea (i.e. the reference provided), the role of lymphangiogenesis is not so clear in other organs (see e.g. PMID: 26485284). Please make sure that the text reflects this ambiguity.

We modified text as follows:

“A less understood facet of neutrophil interactions with the lymphatic system is their ability to promote inflammatory lymphangiogenesis. Lymphangiogeneis has been associated with transplant rejection (Cursiefen, Cao et al. 2004, Kerjaschki, Huttary et al. 2006) and inhibiting this process improves graft survival in corneal transplantation (Amouzegar, Chauhan et al. 2016). However, induction of therapeutic lymphangiogenesis mitigates allograft rejection in a orthotopic model of lung transplantation (Cui, Liu et al. 2015). In addition, preclinical studies in models of inflammatory skin disease (Kajiya, Hirakawa et al. 2006, Kataru, Jung et al. 2009) demonstrated that promoting lymphangiogenesis may contribute to resolution of inflammation”.”

“The functional consequences of inflammation-induced lymphangiogenesis require further investigation. In some settings, expansion of the lymphatic network may support inflammation by promoting transport of immune cells and antigens to the lymph node and enhancing the immune response. In other settings,  lymphatic vessel expansion contributes to resolution of inflammation by draining inflammatory mediators and cells from the site of inflammation (Schwager and Detmar 2019). For instance, inhibiting the lymphatic vasculature led to increased inflammation in mouse models of skin inflammation (Kajiya and Detmar 2006, Kataru, Jung et al. 2009, Huggenberger, Ullmann et al. 2010), inflammatory bowel disease (Jurisic, Sundberg et al. 2013, D'Alessio, Correale et al. 2014) and rheumatoid arthritis (Guo, Zhou et al. 2009). However, inhibiting this process improves graft survival in corneal transplantation (Amouzegar, Chauhan et al. 2016, Hou, Bock et al. 2021)’.

  1. General comment on references: if you are citing more than one study, please fuse the references (should be within one bracket). In the references section: please provide complete author lists (not …et al.), including the final author.

We revised our references.

  1. General comment on abbreviations: introduce once and use consistently (e.g. high endothelial venules, MMP-9)

We apologise for the oversight. We have now fixed the abbreviations.

  1. Several reviews that are cited are very old and should be replaced (or accompanied) by more up-to-date references. E.g. references 28 (line 96), 56 (line 213), 95 (line 364).

New references were added as suggested:

1) Recent studies have identified signals which promote the formation of the lymphatic vasculature (Coso, Bovay et al. 2014, Bálint and Jakus 2021).

  • Coso, S., E. Bovay and T. V. Petrova (2014). Pressing the right buttons: signaling in lymphangiogenesis. Blood. 123: 2614-2624
  • Bálint, L. and Z. Jakus (2021). Mechanosensation and Mechanotransduction by Lymphatic Endothelial Cells Act as Important Regulators of Lymphatic Development and Function. International Journal of Molecular Sciences. 22:

2) On the other hand, if a pathogen is able to survive within neutrophils, then neutrophils can become “trojan horses” concealing pathogens from other immune cells and transporting them from the periphery to secondary lymphoid organs (Appelberg 2007, Coombes, Charsar et al. 2013).

  • Coombes, J. L., B. A. Charsar, S.-J. Han, J. Halkias, S. W. Chan, A. A. Koshy, B. Striepen and E. A. Robey (2013). Motile invaded neutrophils in the small intestine of Toxoplasma gondii-infected mice reveal a potential mechanism for parasite spread. Proceedings of the National Academy of Sciences. 110: E1913-E1922

3) Inhibiting leukocyte trafficking is an effective strategy for treating a myriad of inflammatory diseases (Arseneau and Cominelli 2015, Soehnlein and Libby 2021).\

  • Arseneau, K. and F. Cominelli (2015). Targeting Leukocyte Trafficking for the Treatment of Inflammatory Bowel Disease. Clinical Pharmacology & Therapeutics. 97: 22-28.
  • Soehnlein, O. and P. Libby (2021). Targeting inflammation in atherosclerosis — from experimental insights to the clinic. Nature Reviews Drug Discovery